# Evaluation of Some Safety Parameters of Dual Histamine H_3_ and Sigma-2 Receptor Ligands with Anti-Obesity Potential

**DOI:** 10.3390/ijms24087499

**Published:** 2023-04-19

**Authors:** Kamil Mika, Małgorzata Szafarz, Marek Bednarski, Agata Siwek, Katarzyna Szczepańska, Katarzyna Kieć-Kononowicz, Magdalena Kotańska

**Affiliations:** 1Department of Pharmacological Screening, Jagiellonian University Medical College, Medyczna 9, 30-688 Cracow, Poland; 2Department of Pharmacokinetics and Physical Pharmacy, Jagiellonian University Medical College, Medyczna 9, 30-688 Cracow, Poland; 3Department of Pharmacobiology, Jagiellonian University Medical College, Medyczna 9, 30-688 Cracow, Poland; 4Department of Technology and Biotechnology of Drugs, Faculty of Pharmacy, Jagiellonian University Medical College, Medyczna 9, 30-688 Cracow, Poland; 5Department of Medicinal Chemistry, Maj Institute of Pharmacology Polish Academy of Sciences, Smętna 12, 31-343 Cracow, Poland

**Keywords:** histamine H_3_ receptor antagonists, sigma-2 receptor ligands, obesity, safety, hERG, locomotor activity, liver enzymes

## Abstract

Many studies have shown the high efficacy of histamine H_3_ receptor ligands in preventing weight gain. In addition to evaluating the efficacy of future drug candidates, it is very important to assess their safety profile, which is established through numerous tests and preclinical studies. The purpose of the present study was to evaluate the safety of histamine H_3_/sigma-2 receptor ligands by assessing their effects on locomotor activity and motor coordination, as well as on the cardiac function, blood pressure, and plasma activity of certain cellular enzymes. The ligands tested at a dose of 10 mg/kg b.w. did not cause changes in locomotor activity (except for KSK-74) and did not affect motor coordination. Significant reductions in blood pressure were observed after the administration of compounds KSK-63, KSK-73, and KSK-74, which seems logically related to the increased effect of histamine. Although the results of in vitro studies suggest that the tested ligands can block the human ether-a-go-go-related gene (hERG) potassium channels, they did not affect cardiac parameters in vivo. It should be noted that repeated administration of the tested compounds prevented an increase in the activity of alanine aminotransferase (AlaT) and gamma-glutamyl transpeptidases (gGT) observed in the control animals fed a palatable diet. The obtained results show that the ligands selected for this research are not only effective in preventing weight gain but also demonstrate safety in relation to the evaluated parameters, allowing the compounds to proceed to the next stages of research.

## 1. Introduction

Histamine plays a significant role in numerous physiological functions, such as sleep and wakefulness, circadian rhythm regulation, cognitive function, locomotor activity, food intake, anxiety-like behaviors, and seizures. These functions are mediated through four histamine receptor subtypes that belong to the G-protein-coupled receptor superfamily [1]. Histamine H_3_ receptor activation inhibits histaminergic neurotransmission by reducing the firing of histaminergic neurons in the tuberomammillary nucleus and by inhibiting histamine synthesis and release [2]. This receptor also modulates the release of other neurotransmitters, including acetylcholine, serotonin, glutamate, and GABA by acting as a heteroreceptor on different neurons [3].

Studies have shown that obesity is becoming a global problem that predominantly affects highly developed countries. It is primarily associated with increased access to high-calorie foods and decreased physical activity [4]. In the last two years, during the COVID-19 pandemic, we have seen that many people have been forced to work remotely, which has intensified sedentary lifestyles without physical activity and caused a marked increase in obesity [5]. The prevalence of obesity is estimated to have tripled since 1975 [6,7]. One of the most common treatments for this condition is lifestyle modification, but this method is sometimes ineffective in both the degree and permanence of weight loss. Another method to combat this condition is bariatric surgery, which involves making changes to the gastrointestinal tract to improve the altered mechanisms of neurohormonal regulation of food intake, resulting in a reduction in excess body weight. However, despite the high efficacy of this method, there is great concern about perioperative mortality or postoperative complications [8]. An alternative to these methods may be the use of pharmacological agents, which could safely and effectively reduce body weight in people suffering from obesity. Consequently, many research groups have been searching for many years for effective pharmacotherapeutic strategies to combat obesity. In fact, the last two decades have allowed the discovery of molecular targets responsible for weight control [9]. Despite the approval of numerous drugs by the US Food and Drug Administration (FDA) or/and European Medicines Agency (EMA), most of them were later withdrawn due to the emergence of numerous side effects [10].

Thyroid hormones were the first to be used in the treatment of obesity due to the misconception that increased body fat is associated with hypothyroidism. In fact, thyroid hormones have been shown to cause weight loss, but they mainly lead to the loss of lean body tissue. Side effects, including tachycardia or cardiac arrhythmias, are important limitations of the use of thyroid hormones in the treatment of obesity [11].

In the 1930s, 2,4-dinitrophenol was introduced for the treatment of obesity, with the main purpose of increasing metabolism. Like thyroid hormones, it was withdrawn from treatment due to the compound’s severe toxicity, including agranulocytosis, hepatotoxicity, and even death [12].

An important group of drugs used to treat obesity were sympathomimetic compounds, which were very popular in the 1950s and 1960s. From this group, we should mention fenfluramine, dexfenfluramine, diethylpropion, and phentermine. However, the prevalence of primary pulmonary hypertension, valvular heart disease, and addiction tendencies observed among patients led regulators in some countries to withdraw these drugs from the market [13].

Until a decade ago, sibutramine, a norepinephrine and serotonin reuptake inhibitor, was used to treat obesity. Despite its high efficacy, after leading to numerous cardiovascular complications, it was withdrawn from treatment. Side effects mainly included tachycardia, increased systolic and diastolic blood pressure, or heart palpitations. The use of sibutramine has also been shown to prolong the QT interval [14].

Rimonabant, a CB_1_ cannabinoid receptor antagonist/reverse agonist, has shown efficacy in weight reduction in both animal and human clinical trials [15]. The compound was approved by the EMA for the treatment of obesity, but it was withdrawn after only two years due to mood disorders leading to suicide that occurred after its use [16].

Lorcaserin, a selective serotonin 5-HT_2C_ receptor agonist, is an appetite suppressant. It was approved by the FDA in 2012 for chronic weight management in obese adults [17]. Based on the results of a large follow-up safety study [18], lorcaserin appeared to be largely devoid of cardiovascular toxicity. However, eight years after FDA approval, it was withdrawn from the market when a safety study suggested an increased incidence of cancer among patients treated with this drug [19].

Despite the tumultuous history surrounding the development of therapies for obesity, work continues in the search for new safe and effective drugs. Potential drugs must undergo a series of preclinical studies before entering clinical trials, and preliminary safety studies are crucial in order to eliminate products with significant side effects from further development [20].

In our previous work, we focused primarily on evaluating the efficacy of the tested compounds in the animal model of excessive eating and determining their effects on selected metabolic parameters [21,22,23,24]. The present study focusses primarily on determining the safety (impact on the central nervous system and cardiovascular system) of some of the most active dual ligands of the histamine H_3_ receptor and the sigma-2 receptors. Recent studies have shown that some evaluated histamine H_3_ receptor antagonists possess a nanomolar affinity for sigma-2 receptor binding sites, suggesting that this property might play a role in their overall efficacy and side effects [23,25]. Pitolisant, a drug used in narcolepsy, is a good example of a ligand for both the histamine H_3_ and sigma-2 receptors. It binds to the human histamine H_3_ receptor and sigma-2 receptors with a K_i_ of 1.5 and 6.5 nM, respectively [26]. Similarly, our tested compounds are also dual ligands; therefore, not all of the effects observed after their administration to animals can be attributed to histaminergic transmission.

## 2. Results

### 2.1. The Effect of KSK-61, KSK-63, KSK-73, and KSK-74 on Locomotor Activity in Mice

KSK-74 administered intraperitoneally (i.p.) at a dose of 10 mg/kg b.w. significantly increased locomotor activity in mice compared with that determined in the control group (t = 4.161, df = 14). The other compounds given at the same dose did not have a significant effect on locomotor activity. The results are shown in Figure 1a.

A significant decrease in locomotor activity, already after a single administration, was observed in mice that received KSK-61, KSK-63 or KSK-73 at a dose of 30 mg/kg b.w. compared with control mice that received only vehicle (t = 3.904, df = 12; t = 13.13, df = 14; and t = 5.003, df = 14). No significant changes in locomotor activity were observed in mice that received KSK-74 at the same dose compared with the control group. The results are shown in Figure 1b.

### 2.2. The Effect of KSK-61, KSK-63, KSK-73, and KSK-74 on Motor Coordination in the Chimney Test

The tested compounds did not significantly affect motor coordination. Results were calculated with the Fisher exact probability test and are presented in Table 1. KSK-61, the most notable compound in this test, decreases the ability to exit the chimney at all doses tested in one or two mice per group. Figure 2 shows the exact results of the chimney exit time for individual mice.

### 2.3. Influence of KSK-61, KSK-63, KSK-73, and KSK-74 on Blood Pressure in Normotensive Rats

The compound KSK-61 after single i.p. administration at a dose of 10 mg/kg b.w. did not affect blood pressure. There were no significant differences compared with blood pressure values observed in control animals that received vehicles (1% Tween 80). The results are shown in Figure 3.

After a single i.p. administration of KSK-63 at a dose of 10 mg/kg b.w. to normotensive rats, a significantly increased systolic blood pressure (approx. 16–22 mmHg) was measured 5, 10, and 20 min after administration. The results are shown in Figure 4.

After a single i.p. administration of KSK-73 at a dose of 10 mg/kg b.w. to normotensive rats, significantly decreased systolic (by approx. 12 mmHg) and diastolic (by approx. 10–14 mmHg) blood pressures were observed 30–60 min after administration. The results are shown in Figure 5.

After a single administration of KSK-74 at a dose of 10 mg/kg i.p. to normotensive rats, significantly decreased systolic (by approx. 12 mmHg) and diastolic (by approx. 10–13 mmHg) blood pressures were observed 30–60 min after administration. The results are shown in Figure 6.

### 2.4. Potency to Block the Human Ether-A-Go-Go-Related Gene (hERG)

We established the potency of tested compounds to block the human hERG potassium channel. They were tested in a whole-cell electrophysiological assay in a dose-dependent response and induced inhibition of hERG in the concentration range of 0.570 to 3.397 µM (Table 2). KSK-63 inhibited hERG channel activity close to the reference compound (verapamil); however, the affinity was about 550 times lower than the affinity to the primary target, which for the histamine H_3_ receptor is 3.12 nM (Table 4). The inhibition of hERG by KSK-73 was four times weaker, by KSK-61 about five times weaker, and by KSK-74 almost seven times weaker compared with verapamil (Table 2). All tested compounds bind to primary targets at nanomolar concentration (Table 4).

### 2.5. Effects of KSK-61, KSK-63, KSK-73, and KSK-74 on Heart Rate and Electrocardiogram (ECG) Intervals

None of the compounds tested had a significant influence on QRS, QT, PQ, or heart rate (Table 3).

### 2.6. Effects of 30-Day Intraperitoneal Administration of KSK-61, KSK-63, KSK-73, or KSK-74 on Plasma Levels of AlaT (Alanine Aminotransferase), AspaT (Aspartate Aminotransferase), gGT (Gamma-Glutamyl Transpeptidases), and ALP (Alkaline Phosphatase) of Rats Fed a Palatable Diet

Significant differences in plasma AlaT activity were found between rats in control groups that received different diets (standard or palatable) and were administered vehicle. AlaT activity in the plasma of animals that received i.p. KSK-63, KSK-73 or KSK-74 at a dose of 10 mg/kg b.w. was lower compared with the control group that received a vehicle and was fed palatable feed (Figure 7a). There were no significant changes in the plasma AspaT activity of the animals in the control groups that received standard or palatable feed. However, a higher AspaT activity was observed in the plasma of animals receiving i.p. KSK-61 or KSK-63 at a dose of 10 mg/kg b.w. relative to both control groups receiving standard or palatable feed (Figure 7b). A higher gGT activity was observed in palatable feed-fed animals compared with control animals that only had access to standard feed. Compared with the palatable feed-fed control group, statistically lower gGT activity was recorded in animals treated with compounds KSK-61, KSK-63 or KSK-73 at a dose of 10 mg/kg b.w. (Figure 7c). There were no significant differences in ALP activity in the plasma of animals between the control groups that received standard or palatable feed. Administration of KSK-73 or KSK-74 at a dose of 10 mg/kg b.w. to rats fed palatable feed resulted in lower plasma ALP activity compared with the group receiving the vehicle and having free access to palatable feed. Animals that received KSK-61 or KSK-63 at a dose of 10 mg/kg b.w. had greater ALP activity compared with the control group fed standard feed (Figure 7d).

## 3. Discussion

In this paper, we screened the safety of histamine H_3_/sigma-2 ligands by evaluating their effects on locomotor activity and motor coordination, as well as heart function, blood pressure, and some plasma activities of cell enzymes.

In previous studies, the change in spontaneous locomotor activity (as assessed by the number of infrared beam breaks along a horizontal plane) was demonstrated to be an excellent preclinical predictor of central nervous system/neurobehavioral effects [27,28].

The histamine H_3_ receptor antagonists can increase locomotor activity by increasing histamine release [29]. Histamine H_3_ receptor blockade affects other neurotransmitters such as acetylcholine, dopamine, and serotonin [1,3] and possibly may affect locomotor activity also in this mechanism. Such a condition—increased activity—may be desirable or undesirable, especially since locomotor stimulation is often associated with anxiety-like behaviors [29].

In the present study, we conducted similar tests assessing the effects on locomotor activity in mice of the selected H_3_ histamine receptor antagonists which significantly reduced body weight in the preliminary pharmacological studies. Interestingly, only one of the tested compounds, KSK-74, and only at a dose of 10 mg/kg b.w., significantly increased the mice’s locomotor activity. At a higher dose, the remaining tested ligands showed a sedative effect, while KSK-74 had no effect on locomotor activity. This may be due to the fact that tested compounds also bind to the sigma-2 receptor. Data from the literature also suggest the possible reduction of locomotor activity by sigma-2 receptor ligands [30] or attenuation of cocaine’s motor stimulatory effects [31]. In our previous studies on the effect of tested ligands on body weight in a model of excessive eating in rats, we showed that spontaneous activity—measured by long-term motility monitoring in home cage conditions—is not significantly affected by any of these ligands when administered once and chronically i.p. at a dose of 10 mg/kg/day [21,22,23,24]. Additional studies are required to indicate the exact mechanism underlying this effect.

The locomotor activity test is also necessary for the interpretation of the effect on motor coordination, which we evaluated on mice in the chimney test. Sedated animals cannot come out of the chimney in the allotted time, not because their coordination is disturbed but because they are too sedated. The chimney test is a widely recognized behavioral paradigm that is used in preclinical studies to evaluate the ability of a given drug to interfere with motor coordination in rodents. Motor impairment is shown by the inability of mice to climb backward in the tube in 1 min.

Looking at the obtained results, we can conclude that sedation observed in the animals at exactly the same time after administration of the tested compounds KSK-63 and KSK-73, at a dose of 30 mg/kg b.w., in which we measured their ability to escape from the chimney, did not affect coordination. However, in some cases, the KSK-61 compound prolonged the time of the animal’s exit from the chimney, both in the sedative dose and in the dose that did not reduce motility, but the calculated probability of this effect was not statistically significant. It should be considered that both reduced locomotor activity and ataxia may be related to the neurotoxic effects of the compounds, which might explain the results obtained for the KSK-61 compound.

An important aspect of safety is the potential cardiotoxicity. The effect of histamine on the cardiovascular system is complex; it lowers blood pressure and can cause tachycardia depending on the plasma concentrations [32]. According to Lorenz, baseline plasma histamine concentrations in young healthy male volunteers were approx. ≤1 ng/mL, and when elevated, were associated with increased heart rate (concentrations of 3–5 ng/mL) or with decreased arterial blood pressure (concentrations of 6–8 ng/mL) [33]. Our research has shown that KSK-73 and KSK-74 compounds can significantly lower blood pressure that could be related to an increased histamine effect. However, it is interesting that another tested compound, KSK-63, significantly increased blood pressure only for 20 min after i.p. administration, and KSK-61 had no statistically significant effect on blood pressure. These results highlight the need for further research to clarify these discrepancies. However, all tested compounds did not significantly affect heart rate.

By blocking potassium hERG channels, the duration of an action potential is prolonged, which can induce the formation of an early subsequent depolarizations and torsades de pointes. The mechanism underlying these effects is related to the inhibition of the rapidly activating delayed rectifier potassium current [34]. Early recognition of potential ability to cause the QT prolongation/torsade de pointes is now an essential component of the drug discovery/development program [35]. Thus, potassium hERG channels have become a primary off-target in drug development because their blockade causes potentially serious side effects [34]. Preclinical investigation of some aryl-piperidinyl ether histamine H_3_ receptor antagonists revealed a strong hERG binding. The variation in biphenyl lipophilicity had clearly shown a strong influence on hERG affinity [36]. The results obtained for our compounds show that although these compounds can inhibit hERG activity, they do not significantly affect cardiac parameters, as demonstrated in in vivo studies in rats. It should be noted that the concentrations at which the tested compounds inhibit hERG activity are much higher compared with the concentrations in which they bind to the primary targets, i.e., histamine H_3_ and sigma-2 receptors.

Liver toxicity is one of the most common reasons for an early termination of the development of drugs in people, but there is only a concordance of 43% between the toxicities seen in rodents and humans. This was assessed by a retrospective study of the toxicity of pharmaceuticals in development [37]. The appearance of intracellular enzymes in body fluids can indicate tissue damage. The current best practice recommendation for the assessment of nonclinical safety is that a minimum of four serum parameters should be used to consider hepatocellular and hepatobiliary injury [38,39]. In our study, we determined the activity of four enzymes after thirty administrations of the tested compounds to rats fed palatable feed. Only AlaT and gGT activities were significantly higher in the group fed palatable feed compared with the group fed standard feed. Contrary to our predictions, the activities of AspaT and ALP did not increase in the plasma of the obese control animals but significantly increased in the plasma of rats treated with KSK-61 and KSK-63.

Serum AlaT and gGT are enzymes that are widely used as biologic markers for liver damage and general liver function. The literature also shows that elevated levels of AlaT and gGT in the serum are predictive factors in the development of diabetes [40,41,42]. Elevated activities of AlaT and gGT in the serum are associated with the development of impaired fasting glucose, and simultaneous elevation of both indicators seems to help predict future impaired fasting glucose or diabetes incidence [43]. Interesting and significant not only for safety reasons but also for activity compensating metabolic disorders induced by a palatable diet in animals, the results show that treatment with all tested compounds prevented the increase in plasma activities of AlaT and gGT.

The study observed a certain regularity that compounds with similar chemical structures have similar effects on selected safety parameters. KSK-73 and KSK-74 have an eight-carbon linker, while they differ in the substituents in the aromatic ring. It was observed that both ligands have similar affinity for both the H_3_ histamine receptor and the sigma-2 receptor. Both KSK-73 and KSK-74 lowered blood pressure, which may even seem beneficial, since obesity is often accompanied by other diseases, including but not limited to high blood pressure. Similarly, both ligands did not increase liver enzymes and thus did not adversely affect the liver.

The earlier test compounds can be prioritized or eliminated based on their relative side effects on the central nervous or cardiovascular systems, the sooner resources can be focused on compounds that are more likely to be used in clinical trials [27]. Therefore, our results are extremely important because they can draw the attention of other researchers looking for effective and safe means of weight loss in a pool of similar ligands (histamine H_3_ receptor antagonists) with respect to safety of use depending on the structure and strength of affinity to the receptors mentioned.

## 4. Materials and Methods

### 4.1. Animals

Blood pressure measurement and ECG were performed on male Wistar rats weighing 200–220 g (six months old). Locomotor activity and chimney tests were performed on adult male Albino Swiss (CD-1) mice weighing 18–22 g (six months old). The animals were obtained from an accredited animal house at the Faculty of Pharmacy, Jagiellonian University Medical College, Krakow, Poland. The experimental groups consisted of 6–8 animals. The animals were housed in constant temperature (22–24 °C) and humidity (40–60%) facilities, exposed to 12:12 h light/dark cycles, and were maintained on a standard pellet diet with tap water available ad libitum. Rats and mice were housed in pairs or 8 animals per home cage, respectively.

Standard diet ( 8% fats, 67% carbohydrates, and 25% proteins) contained 100 g feed—280 kcal. Rats fed a palatable diet had access to a diet consisting of milk chocolate with nuts, cheese, salted peanuts, and 7% condensed milk and simultaneously to standard feed (Labofeed B, Morawski Manufacturer Feed, Kcynia, Poland) for 4 weeks. The feeding was not in any way forced, and rats decided themselves when, what, and how much to eat. A palatable diet contained 100 g of peanuts—612 kcal; 100 mL of condensed milk—131 kcal; 100 g of milk chocolate—529 kcal; and 100 g of cheese—325 kcal.

Animals were randomly selected, and all experiments were carried out between 9:00 and 14:00.

### 4.2. Drugs and Chemicals

The tested compounds KSK-61, KSK-63, KSK-73, and KSK-74 (Table 4) were synthesized in the Department of Technology and Biotechnology of Drugs, Faculty of Pharmacy, Jagiellonian University Medical College, Cracow, Poland. The identity and purity of the final product were assessed by nuclear magnetic resonance (NMR) and liquid chromatography–mass spectrometry (LC-MS) techniques (the minimum purity was greater than 95%) [44]. For safety studies, KSK-61, KSK-63, KSK-73, and KSK-74 were suspended in 1% Tween 80.

Based on our previous studies using various compounds from the same chemical group, the most appropriate starting dose for research was 10 mg/kg b.w. At this dose, the tested compounds were active in preventing the development of selected metabolic disorders. In addition, in the study assessing the effect of investigated compounds on motor coordination and locomotor activity, the dose was increased three times according to the standard dose escalation procedure (logarithmic scale).

Heparin was obtained from Polfa Warszawa S.A. (Warsaw, Poland) and thiopental sodium from Sandoz GmbH (Kundl, Austria).

### 4.3. In Vivo Studies

#### 4.3.1. The Effect of KSK-61, KSK-63, KSK-73, and KSK-74 on Locomotor Activity

Locomotor activity was individually recorded for each animal using specifically designed activity cages made of clear Perspex (40 cm × 40 cm × 31 cm, Activity Cage 7441, Ugo Basile, Italy). The cages were supplied with I.R. horizontal beam emitters connected to a counter that records the light-beam interruptions. The compounds suspensions were administered as i.p. injections 30 min before testing. Control animals received injections of vehicles. Each mouse was placed in a cage for a 30 min habituation period. After that time, the number of breaks in the photobeams was measured for 5 min, which is the same time as the observation period used in other tests [45].

#### 4.3.2. The Effect of KSK-61, KSK-63, KSK-73, and KSK-74 on the Motor Coordination in the Chimney Test

The effect of tested compounds on motor impairment were quantified with the chimney test of Boissier et al. (1960). The mice were trained before the test, and only the animals able to get out of the chimney within 1 min were used at the experimental stage. The selected mice were placed in a 25 cm-long and 2.5 cm in diameter horizontally located tube, which was reversed so that the mice could leave it only by climbing up backward until they reached the other end. The test was performed 30 min after injection (i.p.) of the examined compounds. Motor impairment was indicated by the inability of the animals to climb up backwards in the transparent tube within 60 s [46].

#### 4.3.3. Influence of KSK-61, KSK-63, KSK-73, and KSK-74 on Blood Pressure in Rats

Rats were anesthetized with thiopental (70 mg/kg) via i.p. injection. The left carotid artery was cannulated with polyethylene tubing filled with a heparin solution in saline to facilitate pressure measurements using a PowerLab Apparatus (ADInstruments, Sydney, Australia). After a 20 min stabilization period, the test compounds were administered i.p. at a dose of 10 mg/kg b.w. in a constant volume of 1 mL/kg b.w. Blood pressure was recorded immediately before administration and 5, 10, 15, 20, 30, 40, 50, and 60 min after the administration of the test compound [47].

#### 4.3.4. The Effect of KSK-61, KSK-63, KSK-73, and KSK-74 on Normal Electrocardiogram in Rats

In vivo electrocardiographic measurements were performed using an ASPEL ASCARD B5 apparatus, standard lead II, and a paper speed of 50 mm/s. The study was carried out after a single administration of the test compounds (10 mg/kg i.p.). The ECG was recorded just before the compound administration and 5, 10, 20, 30, 40, 50, 60 min after administration. The following parameters were then calculated and analyzed: duration of PQ, QT, QRS, RR intervals, and heart rate [48].

### 4.4. Biochemical Assays

The study evaluating the activities of alanine aminotransferase (AlaT), aspartate aminotransferase (AspaT), gamma-glutamyl transpeptidases (gGT), and alkaline phosphatase (ALP) was carried out after completing the excessive eating model experiments described in our previous work [21,22,23]. Compounds were administered for twenty-eight days at a dose of 10 mg/kg b.w./day. On the 31st day of the experiment, 20 min after i.p. administration of heparin (5000 units/rat) and thiopental (70 mg/kg b.w.), blood was collected from the left carotid artery and then centrifuged at 600× *g* (15 min, 4 °C) in order to obtain the plasma. To determine the activities of AlaT, AapaT, gGT, and ALP in the plasma samples, a standard enzymatic spectrophotometric test (Biomaxima S.A., Lublin, Poland) was used.

### 4.5. In Vitro Electrophysiological Studies

Electrophysiology experiments were performed on a QPatch16X automatic patch clamp platform (Sophion Bioscience, Ballerup, Denmark). Chinese hamster ovary (CHO) cells stably expressing the human ERG potassium channel (Kv11.1) were cultured using standard procedures. On the day of the experiment, cells were collected from the culture flask using TrypLE Express solution (Life Technologies, Thermo Fisher Scientific, Carlsbad, California, United States) and resuspended in EX-CELL ACF CHO serum-free medium (Merck KGaA, Sigma-Aldrich, Darmstadt, Germany), supplemented with 25 mM HEPES, 100 U/mL penicillin/streptomycin, and 0.04 mg/mL soybean trypsin inhibitor (all components from Sigma Aldrich).

The cells were placed in the lidless microtube and located onboard the automated electrophysiology instrument, where they were automatically relocated to a built-in centrifuge. Cells in suspension were spun down and washed twice in extracellular Ringer’s solution (EC: 2 mM CaCl_2_, 1 mM MgCl_2_, 10 mM HEPES, 4 mM KCl, 145 mM NaCl, 10 mM glucose, pH 7.4, and 306 mOsm). The intracellular buffer (IC) composition was as follows: 5.374 mM CaCl_2_, 1.75 mM MgCl_2_, 31.25 mM KOH, 10 mM EGTA, 10 mM HEPES, 120 mM KCl, 4 mM Na_2_-ATP, pH 7.2, and 284 mOsm.

Cells were transferred to pipetting wells of single-use 16-channel planar patch chip plates (QPlate 16X, Sophion Bioscience, Ballerup, Denmark) and gigaseals were formed upon application of a combined suction/voltage protocol in a standard protocol supplied by Sophion Bioscience for CHO cells. Once the whole-cell mode was obtained, the cells were kept in a holding potential of -90 mV between the programmed stimulation protocols.

Whole-cell potassium currents were measured in response to repeatedly executed voltage protocols constituted of the following steps: brief clamping to −50 mV (200 ms), subsequent depolarization to 20 mV for 4000 ms, and final repolarization to −50 mV when the outward tail current was measured for 4000 ms.

The average current peak values determined in the presence of particular concentrations of the compound tested were subjected to a sigmoidal dose–response curve fitting, where the bottom plateau parameter was constrained to 0. The IC_50_ values were then calculated from the obtained sigmoidal dose–response curves. Data were analyzed using QPatch Assay Software (v5.0, Sophion Bioscience, Ballerup, Denmark) and represent the mean of three experiments carried out on distinct cells.

### 4.6. Statistical Analysis

Statistical calculations were performed using the GraphPad Prism 6 program (San Diego, CA, USA). Results are presented as arithmetic means with a standard deviation or standard error of the mean. The normality of the data sets was determined using the Shapiro–Wilk test. Statistical significance was calculated using the following: unpaired *t*-test (locomotor activity), Mann–Whitney test or Fisher exact probability test (motor coordination), one-way ANOVA, Tukey post hoc (liver enzyme activity), and two-way ANOVA/Bonferroni post hoc test (blood pressure, ECG). Differences were considered statistically significant at * *p* ≤ 0.05, ** *p* ≤ 0.01, and *** *p* ≤ 0.001.

## 5. Conclusions

In summary, in the research presented here, it was shown that the ligands tested at a dose of 10 mg/kg b.w. did not affect locomotor activity (except KSK-74) or significantly impair motor coordination. When evaluating the effects on the cardiovascular system, it was shown that the selected compounds could lower blood pressure, which seems logical given the pharmacological effects of histamine. Despite the ability of the tested ligands to block hERG potassium channels, their effects on the electrocardiogram were not demonstrated in vivo. It is noteworthy that repeated administration of the tested compounds prevented an increase in AlaT and gGT activity observed in the control animals fed palatable feed. Such an effect is important not only for reducing metabolic disorders usually accompanying obesity but also for the further development of these compounds, considering many potential ligands do not undergo clinical trials due to liver toxicity. So far, the results demonstrate that the evaluated compounds are promising and demonstrated safety in relation to the evaluated parameters, allowing the compounds to proceed to the next stages of research.

## Figures and Tables

**Figure 1 ijms-24-07499-f001:**
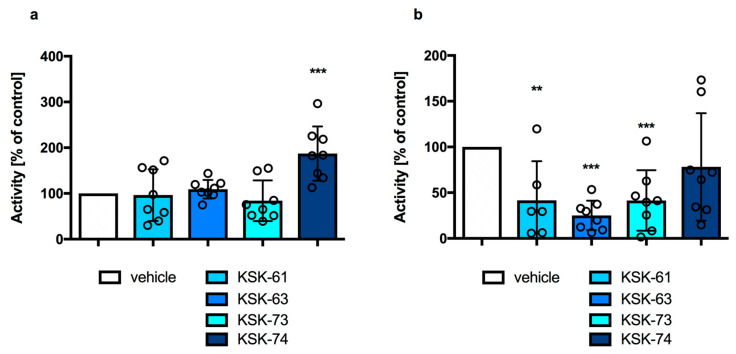
Locomotor activity in mice after a single administration of the tested compounds at a dose of 10 mg/kg b.w. (**a**) or at a dose of 30 mg/kg b.w. (**b**). Results are presented as mean ± SD, *n* = 6–8. Statistical analysis: Shapiro–Wilk normality test and unpaired *t*-test, ** *p* < 0.01; *** *p* < 0.001 vs. vehicle group.

**Figure 2 ijms-24-07499-f002:**
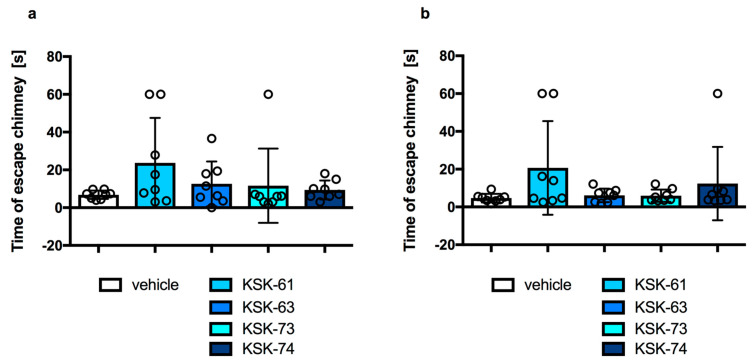
Time of escape from the chimney after a single administration of the tested compounds at a dose of 10 mg/kg b.w. (**a**) or at a dose of 30 mg/kg b.w. (**b**). Mice that did not exit the chimney in 60 s were assigned a score of 60 s. Results are presented as mean ± SD, *n* = 8. Statistical analysis: Shapiro–Wilk normality test and Mann–Whitney test.

**Figure 3 ijms-24-07499-f003:**
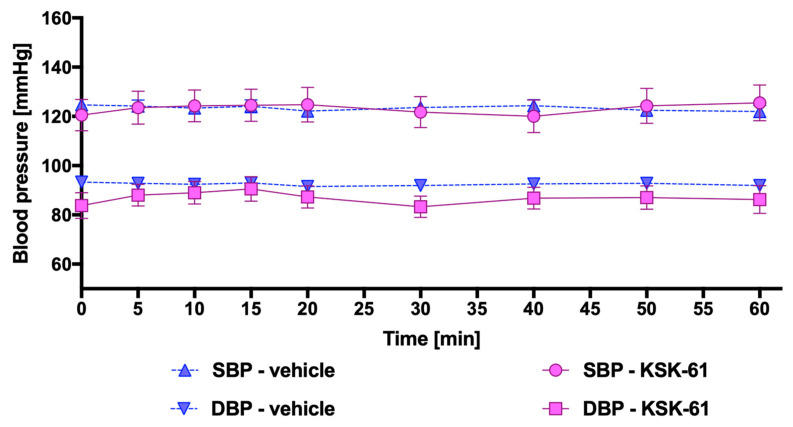
Effect of KSK-61 on blood pressure after a single administration (10 mg/kg i.p.). Results are presented as mean ± SEM, *n* = 6. Statistical analysis: two-way ANOVA with repeated measures, Bonferroni’s post hoc tests. SBP—systolic blood pressure; DBS—diastolic blood pressure.

**Figure 4 ijms-24-07499-f004:**
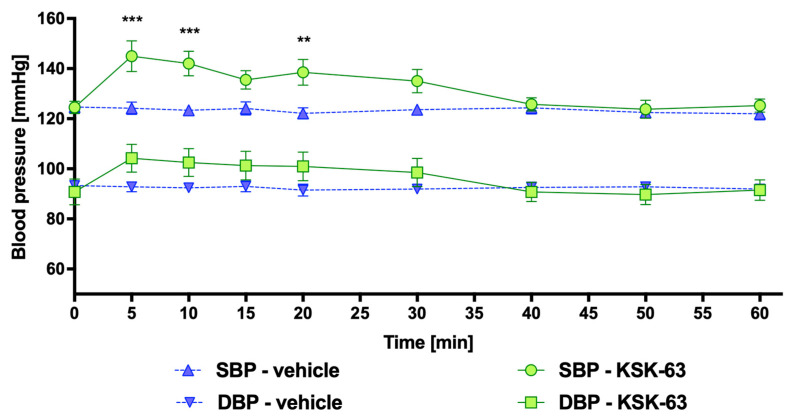
Effect of KSK-63 on blood pressure after a single administration (10 mg/kg i.p.). Results are presented as mean ± SEM, *n* = 6. Statistical analysis: two-way ANOVA with repeated measures, Bonferroni’s post hoc tests, ** *p* < 0.01; *** *p* < 0.001 vs. vehicle group. SBP—systolic blood pressure; DBS—diastolic blood pressure.

**Figure 5 ijms-24-07499-f005:**
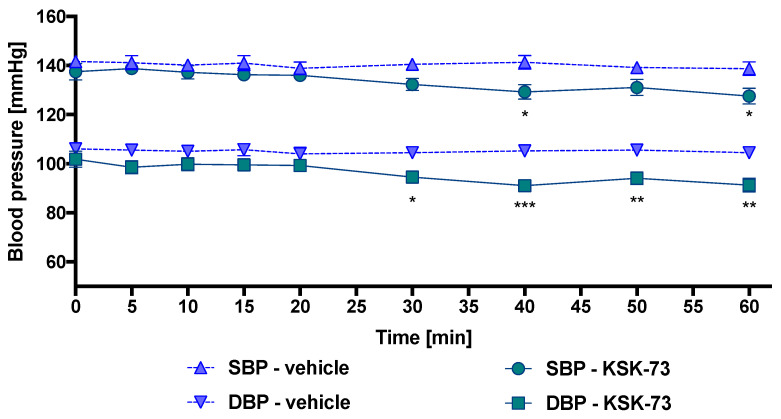
Effect of KSK-73 on blood pressure after a single administration (10 mg/kg i.p.). Results are presented as mean ± SEM, *n* = 6. Statistical analysis: two-way ANOVA with repeated measures, Bonferroni’s post hoc tests, * *p* < 0.05; ** *p* < 0.01; *** *p* < 0.001 vs. vehicle group. SBP—systolic blood pressure; DBS—diastolic blood pressure.

**Figure 6 ijms-24-07499-f006:**
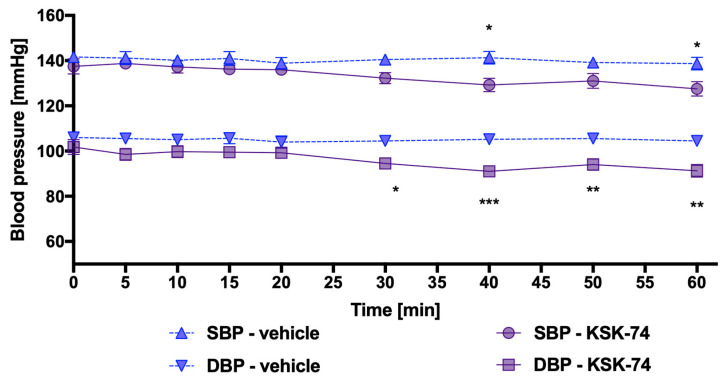
Effect of KSK-74 on blood pressure after a single administration (10 mg/kg i.p.). Results are presented as mean ± SEM, *n* = 6. Statistical analysis: two-way ANOVA with repeated measures, Bonferroni’s post hoc tests, * *p* < 0.05; ** *p* < 0.01; *** *p* < 0.001 vs. vehicle group. SBP—systolic blood pressure; DBS—diastolic blood pressure.

**Figure 7 ijms-24-07499-f007:**
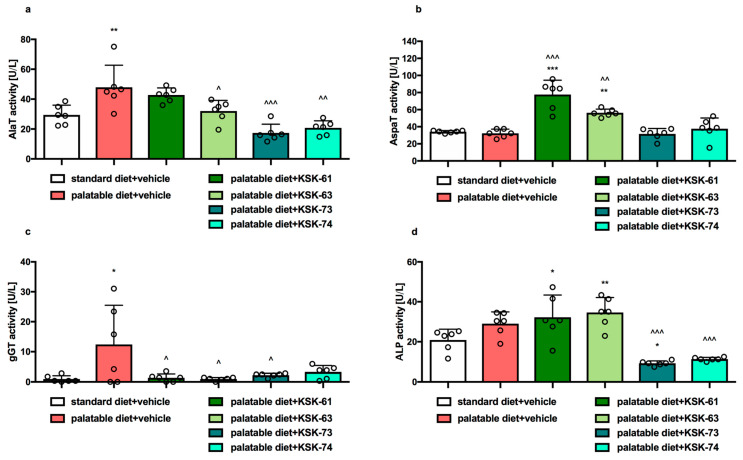
Effect of the tested compounds on plasma levels of (**a**) AlaT activity, (**b**) AspaT activity, (**c**) gGT activity, and (**d**) ALP activity. Results are presented as mean ± SD, *n* = 6. Comparisons were performed by one-way ANOVA and Tukey’s post hoc test. Significant against control rats fed a standard diet; *,^ *p* < 0.05; **,^^ *p* < 0.01; ***,^^^ *p* < 0.001.

**Table 1 ijms-24-07499-t001:** The effect of tested compounds on motor coordination in the chimney test.

Compound (mg/kg)	*n*	% of Mice Impaired
vehicle	8	0%
KSK-61 (10)	8	25%
KSK-61 (20)	8	12.5%
KSK-61 (30)	8	25%
KSK-63 (10)	8	0%
KSK-63 (20)	8	0%
KSK-63 (30)	8	0%
KSK-73 (10)	8	12.5%
KSK-73 (20)	8	0%
KSK-73 (30)	8	0%
KSK-74 (10)	8	0%
KSK-74 (20)	8	12.5%
KSK-74 (30)	8	12.5%

Values shown as percentage of animals with motor impairment (*n* = 8). Compounds were administered intraperitoneally. Statistical analysis of the data was performed using the Fisher exact probability test.

**Table 2 ijms-24-07499-t002:** Inhibition of hERG currents by tested compounds.

Compound	Mean IC_50_ ± SEM (µM)
KSK-61	2.657 ± 0.217
KSK-63	0.570 ± 0.050
KSK-73	2.067 ± 0.402
KSK-74	3.397 ± 0.289
Verapamil	0.525 ± 0.052

**Table 3 ijms-24-07499-t003:** Effects of intraperitoneal injection of compounds tested at a dose of 10 mg/kg on heart rate and ECG intervals.

Parameter		Time (min)				
		0	15	30	45	60
KSK-61 (10 mg/kg b.w.)	Heart rate (beats/min)	373.2 ± 20.3	369.5 ± 9.5	380.2 ± 30.5	349.5 ± 21.3	360.4 ± 20.3
	PQ (ms)	53 ± 1.6	52.8 ± 1.5	54.5 ± 3.5	53.3 ± 1.8	53 ± 1.6
	QRS (ms)	20.8 ± 0.8	21.5 ± 0.5	21.8 ±1.1	21.8 ± 0.8	21.8 ± 0.4
	QT (ms)	56.3 ± 4.1	56.3 ± 2.2	58.8 ± 2.2	57.5 ± 4.3	55 ± 5.0
KSK-63 (10 mg/kg b.w.)	Heart rate (beats/min)	393.8 ± 34.8	362.8 ± 38.0	365.2 ± 34.8	344.3 ± 30.8	344.3 ± 30.8
	PQ (ms)	57.5 ± 2.5	55.0 ± 5.0	58.75 ± 1.3	55.0 ± 5.0	62.5 ± 2.1
	QRS (ms)	20.0 ± 0.0	20.0 ± 0.0	20.0 ± 0.0	20 ± 0.0	20.0 ± 0.0
	QT (ms)	63.3 ± 5.1	64.4 ± 2.6	65.3 ± 4.7	62.1 ± 1.6	65.2 ± 3.6
KSK-73 (10 mg/kg b.w.)	Heart rate (beats/min)	370.0 ± 23.9	375.5 ± 23.9	377.1 ± 27.3	395.8 ± 33.3	377.0 ± 26.4
	PQ (ms)	56.3 ± 2.2	51.3 ± 2.2	51.3 ± 2.2	52.5 ± 2.5	53.8 ± 2.2
	QRS (ms)	22.7 ± 5.4	22.7 ± 5.4	22.7 ± 5.4	20.0 ± 0.0	20.0 ± 0.0
	QT (ms)	61.3 ± 2.2	63.8 ± 4.1	60.0 ± 7.1	58.8 ± 2.2	63.8 ± 6.5
KSK-74 (10 mg/kg b.w.)	Heart rate (beats/min)	381.8 ± 19.9	356.3 ± 18.9	355.3 ± 32.2	364.6 ± 18.1	370.8 ± 23.9
	PQ (ms)	56.3 ± 4.1	53.8 ± 4.1	56.3 ± 2.2	57.5 ± 2.5	55.0 ± 5.0
	QRS (ms)	20.0 ± 0.0	20.0 ± 0.0	20.0 ± 0.0	20.0 ± 0.0	20.0 ± 0.0
	QT (ms)	62.5 ± 4.3	60.0 ± 5.0	61.3 ± 6.5	65.0 ± 6.1	62.5 ± 8.3

Statistical analysis: repeated measures one-way ANOVA (Dunnett’s post hoc); *n* = 6 rats.

**Table 4 ijms-24-07499-t004:** Structures and affinities to primary targets of tested compounds [25].

Symbol	Structure	H_3_R Ki (nM)	σ_1_R Ki (nM)	σ_2_R Ki (nM)	H_3_/σ1	H_3_/σ_2_	σ_1_/σ_2_
KSK-61	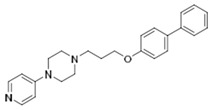	21.1	638.26	108.14	0.03	0.20	5.90
KSK-63	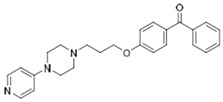	3.12	726.11	29.24	0.004	0.11	24.83
KSK-73	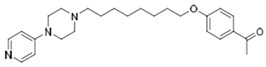	40.5	408.30	59.70	0.1	0.68	6.84
KSK-74	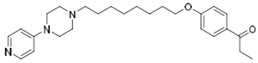	38.9	274.16	65.92	0.14	0.59	4.16

## Data Availability

The data presented in this study are available on request from the corresponding author.

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
