# Peer review of "Evaluation of Some Safety Parameters of Dual Histamine H3 and Sigma-2 Receptor Ligands with Anti-Obesity Potential"

_ijms, 2023, doi:10.3390/ijms24087499_

Round 1
Reviewer 1 Report
This manuscript describes “Safety evaluation of dual histamine H3 and sigma-2 receptor ligands with anti-obesity potential”. The aim and rationale of the manuscript is well. However, the expression of the paper is poor. The following points should be addressed by the authors before accepting this manuscript to be published:
1. The experiment has been completed and the writing tense should be past tense, but there are many present tenses present in the paper. For example, in the line 32, “show”. In the line 128, “are” and so on.
2. There are many grammatical errors in the paper. For example, in the line 71, the singular and plural are incorrectly expressed in the sentence.
in the line 75, comma and “and” are used together in a sentence, in the line 245, “it is possible may affect” is wrong, in the line 267 and so on.
3. Some parts of the text are not clear. For example, in the line 199, “by KSK-73 was four times weaker, by KSK-61 was about five times weaker and by KSK-74 was almost seven times weaker compared to verapamil.”
4. The references are not standardized in some places. For example, the page number of reference 1 is written incorrectly. The year of reference 19, 40 and 42 are not bolded, and the year of reference 44 is written incorrectly.
5. The description of the experiment results is too simple and should be more specific.
6. In the Table 3, the “53±1,6” of the compound KSK-61 is wrong, and the valid digits of the compound KSK-63 is inconsistent. In the Table 4, the number of H3 receptors should be subscripted.
Reviewer 2 Report
Briefly, the article in question evaluates the effect of four H3 and sigma-2 receptor ligands (administered I.P.) on parameters related to toxicity/safety. The article in question has important results and used adequate methodology and is therefore suitable for publication. Still, here are some suggestions:
General suggestions:
• It would be interesting to think about modifying the structure of the article, especially in relation to the order of sections, which would be better this way: introduction, objectives, material and method, results, discussion and conclusions;
• The term H3 receptor ligand is used extensively throughout the article. This term gives the impression of agonist activity but reading some of the cited references, especially reference 21, it seems to me that the compounds tested have an H3 antagonistic action, thus increasing histaminergic transmission. Thus, I suggest specifying, starting with the title, the type of activity (agonist, antagonist or inverse agonist) that the tested compounds have on the receptor as this was only specified in the discussion.
• The doses used were different in some of the “in vivo” tests. It would be interesting to make explicit the criteria used to choose the doses.
Suggestions for specific sections:
Title:
· Safety involves a wide spectrum of parameters. As not all safety parameters were evaluated, I suggest changing the title to: “Evaluation of some safety parameters of dual histamine H3....”
Abstract:
• Adequate but see that the verb tenses must be in the past, especially the objectives.
• Abbreviations for hERG, ALAt and gGT appear for the first time and must have their meaning spelled out in parentheses.
Introduction:
• Adequate but part of the introduction, especially the last paragraph is more related to the discussion or even the conclusions. I suggest putting the information from the last paragraph in the conclusions item or at the end of the discussion.
Objective:
I suggest putting a separate section on objectives after the introduction section.
Material and methods:
• Very detailed but I suggest relocating the material and methods section before the results section;
• Even using previous studies as a basis (references 21, 22 and 23), it would be interesting, if there is still space, a brief description of the difference between standard and palatable diets.
• Animals:
Although the meaning is obvious, the abbreviation ECG appears for the first time and therefore its meaning needs to be in parentheses (in full);
It would be interesting to describe how many animals were packed in each box.
• Drugs and Chemicals: The meanings of the abbreviations NMR and LC-MS need to be enclosed in parentheses (in full).
• In vitro electrophysiology studies: The cell culture preparation is well detailed but: What is CHO? Is it a cell lineage? This could be described.
• Biochemical analysis: the procedure is well described but it would be interesting to list the doses used.
Results:
• They are adequate and well described, but in the description of Figure 7, the terms standard feed and palatable feed appear for the first time, which should be described in the method.
Discussion:
· It is well structured but also lacks a focus on the physiological role of the sigma-2 receptor, including endogenous ligands. Think of doing as was done with the H3 receptor, for which the role of histaminergic transmission was associated, and doing the same for the sigma-2 receptor.
Conclusions:
• For the most part, the conclusions are adequate and in agreement with the results. However, as not all parameters related to the safety of use were evaluated, I think it is premature to say that the compounds are safe to proceed to the other stages of development or perhaps it would be better to specify which stages would be the next ones. For example, parameters related to acute and chronic toxicity are missing and, in addition, the effects of compounds on locomotion, blood pressure and ECG were evaluated in a single dose or for a short period. Thus, I only suggest concluding that “so far, the results demonstrate that the evaluated compounds are promising and demonstrated safety in relation to the evaluated parameters, but further studies are needed to assess safety” or else “so far, the results demonstrate that the evaluated compounds are promising and demonstrated safety in relation to the evaluated parameters, allowing the compounds to be evaluated later in the stages.........”
References:
• Adequate
Round 2
Reviewer 1 Report
The manuscript can be accepted in present form.